# Immune Checkpoint Inhibitors in the Treatment of Renal Cancer: Current State and Future Perspective

**DOI:** 10.3390/ijms21134691

**Published:** 2020-06-30

**Authors:** Daniele Lavacchi, Elisa Pellegrini, Valeria Emma Palmieri, Laura Doni, Marinella Micol Mela, Fabrizio Di Maida, Amedeo Amedei, Serena Pillozzi, Marco Carini, Lorenzo Antonuzzo

**Affiliations:** 1Clinical Oncology Unit, AOU Careggi, 50134 Firenze, Italy; daniele.lavacchi@yahoo.it (D.L.); pellegrinie81@gmail.com (E.P.); valeriaemmap@gmail.com (V.E.P.); doni.laura@gmail.com (L.D.); micolmela@yahoo.it (M.M.M.); fabridima90@gmail.com (F.D.M.); serena.pillozzi@unifi.it (S.P.); marco.carini@unifi.it (M.C.); 2Department of Experimental and Clinical Medicine, University of Firenze, 50134 Firenze, Italy; amedeo.amedei@unifi.it

**Keywords:** immune checkpoint inhibitors, renal cell carcinoma, tyrosine kinase inhibitors, predictive biomarkers

## Abstract

Systemic treatment of renal cancer (RCC) has undergone remarkable changes over the past 20 years with the introduction of immunotherapeutic agents targeting programmed cell death (PD-1)/programmed death-ligand 1 (PD-L1) axis, as a single-agent or combined with anti-CTLA-4 monoclonal antibodies (MoAbs) or a multi-target vascular endothelial growth factor-(VEGF) tyrosine kinase inhibitor (TKI). In this paper, we review the main evidence on the use of Immune Checkpoint Inhibitors (ICIs) for RCC treatment from the first demonstration of activity of a nivolumab single agent in a phase I trial to the novel combination strategies (anti-PD-1 plus anti-CTLA4 or anti-PD-1 plus TKI). In addition, we discuss the use of anti-PD-1/PD-L1 agents in patients with non-clear cells and rare histological subtype RCC. Then, we critically examine the current findings in biomarkers that have been proposed to be prognostic or predictive to the response of immunotherapy including immune gene expression signature, B7-H1 expression, PBRM1 loss of function, PD-L1 expression, frame shift indel count, mutations in bromodomain-containing genes in patients with MiT family translocation RCC (tRCC), high expression of the T-effector gene signature, and a high myeloid inflammation gene expression pattern. To date, a single biomarker as a predictor of response has not been established. Since the dynamic behavior of the immune response and the different impact of ICI treatment on patients with specific RCC subtypes, the integration of multiple biomarkers and further validation in clinical trials are needed.

## 1. Introduction

Renal cell carcinoma (RCC) is the third urological cancer accounting for approximately 3–5% of all newly diagnosed tumors [1]. Around 400,000 cases of RCC are diagnosed worldwide every year with 30% of patients having advanced-stage or metastatic disease at diagnosis with an estimated 5-year survival rate of 10% [1,2]. Among histological subtypes, clear cell (cc) histology is most prevalent, accounting for about 75% of cases and the other histologies mainly encompass papillary (20%) and chromophobe RCC (5%). The other entities are very rare including translocation-associated RCC, medullar RCC, and collecting duct carcinoma.

Significant changes in the treatment landscape for patients with RCC have been registered over the past 20 years [3]. Traditional immunotherapy involving IFN-α and IL-2 was the main treatment for metastatic RCC (mRCC) before the approval of targeted therapies. The efficacy of IFN-α for mRCC patients was first reported in 1989 by Oliver et al. [4] with a response rate of roughly 15% and with 6 monthes increase in overall survival (OS). According to the Cochrane meta-analysis, IFN-α offers only a small survival improvement when compared to other control regimens. Moreover, responses to IFN-α were not long-lasting and few patients showed complete responses (CRs) [5]. High dose (HD) IL-2 was approved in 1992 for treating mRCC, which offered an objective response rate (ORR) of 10–20% and a complete remission in 7–10% of patients. Nearly all patients remained free of disease up to 20 years after therapy [6]. Therefore HD IL-2 had become the preferred treatment for mRCC, but severe toxicity in various organ systems, most significantly the heart, lungs, kidney, and central nervous system, limited its applicability to highly selected patients [7,8]. Since the evidence of activity of targeted therapy as the multitarget vascular endothelial growth factor (VEGF) tyrosine kinase inhibitors (TKIs) and mammalian target of rapamycin (mTOR) inhibitors, the treatment of cytokine alone has gradually fallen out of favor.

The treatment options for patients with RCC have been expanding in the last few years [9] after the advent of immunotherapeutic agents targeting programmed death-1 (PD-1)/programmed death-ligand 1 (PD-L1) axis as a single-agent (Table 1) or combined with anti-CTLA-4 monoclonal antibodies (MoAbs) [10]. More recently, the combination of immunotherapeutic agents with antiangiogenic agents has proven to be a promising therapeutic strategy [11].

In this case, we review the main evidence on the use of ICIs for RCC treatment. Then we critically examine the current findings in biomarkers that have been proposed to be predictive of response to immunotherapy.

## 2. Methodology

We used the PubMed general portal to search for prior reviews that match our study question (immunotherapy, renal carcinoma) and then for original published/unpublished articles (61 full text articles assessed for eligibility). Additionally, we searched on ClinicalTrials.gov for clinical studies. 

## 3. Anti-PD-1/PD-L1 Single Agent

The activity of nivolumab was first investigated in a phase I study [12] that evaluated the safety, pharmacokinetics, and efficacy in some solid tumors, including RCC. A maximum tolerated drug dose has not been defined in this study (investigated dose: up to 10 mg per kilogram of body weight every two weeks). Responses were observed in 4 out of 17 (24%) RCC patients treated with a dose of 1 mg per kilogram and in 5 out of 16 (31%) treated with 10 mg per kilogram. The toxicity profile was acceptable, drug-related grade (G) 3 or 4 adverse events (AE) that occurred in 14% of patients (Table 2).

In a blinded, randomized, multi-center phase II trial, mRCC patients who had received at least one previous anti-angiogenic therapy received nivolumab 0.3 mg/kg, 2 mg/kg, or 10 mg/kg every three weeks [13]. The primary end point was to compare the Progression-Free Survival (PFS) in the three treatment arms to evaluate a possible correlation with the administered dose. Secondary end points were PFS, Objective Response Rate (ORR), time to response, duration of response, Overall Survival (OS) rate, and Adverse Event (AE) rate. The 168 randomized patients were stratified by the Motzer Score For Renal Cell Carcinoma(MSKCC) risk group and number of previous treatments in the metastatic setting. Median PFS was 2.7 months (80% CI, 1.9 months to 3.0 months), 4.0 months (80% CI, 2.8 months to 4.2 months), and 4.2 months (80% CI, 2.8 months to 5.5 months) in the 0.3 mg/kg, 2 mg/kg, and 10 mg/kg group, respectively (stratified trend test *p* = 0.9). ORR was 20%, 22%, and 20% in each arm (exact Cochran-Armitage trend test *p* = 1.0) and median OS were 18.2 months, 25.5 months, and 24.7 months, respectively. No association between dose and response was observed. Treatment-related AEs were predominantly low grade with G3–4 AEs in 11% of cases. No high-grade pneumonitis was observed. In an exploratory analysis, median PFS was 4.9 months in the PD-L1 ≥ 5% subgroup and 2.9 months in the PD-L1 < 5% subgroup. 

A randomized, multi-center, open-label, phase III trial, Checkmate 025, investigated the effectiveness of nivolumab vs. everolimus (mTOR inhibitor) [14]. The study population included mRCC patients with a clear-cell component previously treated with one or two anti-angiogenics. The primary end point was OS, which was significantly longer in the nivolumab arm (25 months) than in the everolimus arm (19.6 months). Secondary end points included ORR and safety. The nivolumab advantage (compared with everolimus) was also evident in secondary end points with an ORR of 25% vs. 5% (*p* < 0.001) and a better safety profile. In particular, G3 or G4 treatment-related AEs occurred in 19% and 37% of patients in the nivolumab arm and in the everolimus arm, respectively. Most commonly reported AEs were fatigue (3%) with nivolumab and anemia (8%) with everolimus. Median PFS were 4.6 and 4.4 months, respectively (*p* = 0.11). In a post-hoc analysis of patients who had not progressed or died at 6 months, median PFS was 15.6 months in the nivolumab group and 11.7 months in the everolimus group. The expression of PD-L1 was not associated with response to nivolumab. Assuming 1% and 5% as cut-off, a correlation between PD-L1 expression and poor prognosis was reported (likely due to the type of tumor and histology). An update has been recently presented after more than 5 years of follow-up [15]. ORR (23% with nivolumab vs. 4% with everolimus) and OS (25.8 months vs. 19.7 months) remained superior with nivolumab and 28% of responses to nivolumab were ongoing, while, with everolimus, ongoing responses were observed in 18% of patients. Furthermore, in the nivolumab group, median duration of the response (DOR) was longer (18.2 months vs. 14.0 months) than in the everolimus group. 

A phase II trial explored the use of intermittent nivolumab in mRCC patients that had received prior anti-angiogenic therapy [16]. Patients were treated with nivolumab for 12 weeks and those who achieved ≥10% reduction in the tumor burden initiated a treatment-free observation phase. The primary objective was feasibility of intermittent nivolumab, as the percentage of patients eligible for intermittent therapy who have accepted this treatment scheme (assumed as “feasible” if the acceptance rate was ≥80%). A total of 14 patients were enrolled and 5 patients accepted intermittent nivolumab treatment. With a median follow-up of 48 weeks, only one patient needed to restart therapy while the others kept their response for a median of 34 (range, 16–53) weeks off therapy. It was a small-sized study, which brings out the following concepts: in the era of combination therapies, identifying clear responder patients to immunotherapy who could benefit from a suspension, may have an important task in terms of reducing toxicity and costs. 

A phase II study investigated the switch to nivolumab vs. TKI continuation after 12 weeks of TKI induction therapy [17]. This trial was prematurely closed because of low accrual rate. It included patients who had an advanced or metastatic clear cell RCC with partial response (PR) or stable disease (SD) to induction therapy with sunitinib or pazopanib. Patients were randomized to TKI continuation or switched to nivolumab (240 mg intravenously [IV] every two weeks or 480 mg IV every four weeks). Stratification factors were MSKCC risk, previous TKI, and response to TKI. In a further interim analysis, ORR was 64% in the nivolumab group vs. 70% in the TKI group when assessed from the start of induction therapy (*p* = 0.76) and 16% vs. 48%, respectively, when assessed from the time of randomization (*p* = 0.032). PFS from randomization was 3.0 months for nivolumab and 11.9 months for TKI (*p* = 0.0026). At a median follow-up of 12.9 months, median OS was not reached. The continuation of TKI in patients sensitive to these drugs appeared to be more effective than the switch to nivolumab. However, the small sample size and the early study termination may have limited the strength of the results. In the last few years, single-agent ICIs seem to be promising in the perioperative setting and may change the usual clinical practice.

A randomized, unblinded phase III trial comparing perioperative nivolumab vs. observation in RCC patients undergoing nephrectomy is ongoing [18]. The rationale for this study was to prime the immune system before nephrectomy by taking into account the feasibility of immunotherapy single-agent and its good safety profile that does not delay surgery. Patients with clinical stage ≥ T2N × M0 or TanyN + M0 or oligometastatic (≤3 sites of metastases that can be radically resected) are randomized to receive nivolumab 480 mg every four weeks of the schedule for only one neoadjuvant administration before nephrectomy, which is followed by nine adjuvant cycles vs. observation. The primary end point is recurrence-free survival (RFS). The study strength is the opportunity to analyze tissue before and after exposure to immunotherapy and, consequently, to study potential markers. 

Another ongoing study (phase II single center) is evaluating nivolumab in the perioperative setting in metastatic patients who plan to undergo cytoreductive nephrectomy [19]. Nivolumab is administered at a dose of 3 mg/kg every 2 weeks for eight weeks pre-cytoreductive nephrectomy and post-operatively until undergoing progressive disease (PD). The primary end point is represented by safety while secondary end points are represented by a response evaluation and an exploratory biomarker analysis. At median follow-up of 12.5 months, an acceptable toxicity profile was registered. The baseline immune gene expression pattern was different between responders and non-responders. In particular, the presence of tumor infiltrating lymphocytes was associated with lasting responses.

Another ICI, pembrolizumab, has been studied in a single-arm, open-label, phase II study as first-line treatment in patients with RCC. This study had two cohorts: patients with advanced clear cell RCC and non-clear cell RCC (nccRCC). The primary end point was ORR. Results of cohort A were presented at a median follow-up of 23 months. The ORR was 36.4% with a CR of 2.7%, a time to respond of 2.8 months, and a median duration of response was not reached. Median PFS was 7.1 months and the median OS was unreached. Moreover, considering a combined positive score (CPS), ORR in patients with CPS ≥ 1 was 44.2%, and in patients with CPS < 1 was 29.3% [20].

Results of cohort B were presented at a median follow-up of 15 months. The predominant histology was papillary (72%), which was followed by the chromophobe (13%) and unclassified (16%). Moreover, 62% were PD-L1+. In RC patients with the non-clear cell variant, the ORR was 26.1% while the median duration of response was 15.3 months. In particular, ORR was 28.0% in papillary, 9.5% in chromophobe, and 30.8% in unclassified nccRCC. ORR in patients with CPS ≥ 1 was 35.3% and in patients with CPS < 1 was 10.3% [21].

Pembrolizumab has been studied in the adjuvant setting. A randomized, double-blind, placebo-controlled, phase III study is underway (KEYNOTE-564 trial) [22]. The primary end point is disease free survival (DFS) per the investigator assessment and the secondary end point is OS. Randomization will be stratified by the metastasis stage (M0 vs. M1).

In the same setting, another study evaluating the activity of atezolizumab is ongoing [23]. RCC patients who have undergone nephrectomy and are at high risk of recurrence (T2 grade 4, T3a grade 3–4, T3b/c any grade, T4 any grade, or TxN+ any grade) or have had complete resection of limited metachronous/synchronous metastasis will be randomized to receive atezolizumab 1200 mg IV every three weeks or placebo IV every three weeks for 16 cycles or one year. Stratification factors are disease stage, geographic region, and PD-L1 status on tumor-infiltrating immune cells. The primary end point is independent of review facility-assessed DFS.

Avelumab, another ICI, has also been shown to be effective as monotherapy in first and second line treatment, according to phase 1b results from the Javelin Solid Tumor Trial, in which Disease Control Rate DCR was similar in both groups (77.4% in the first line, 75.0% in the second line) and a 12-month OS rates was 83.7% and 65.0%, respectively, in the first and second line [24].

## 4. Anti-PD-1/PD-L1 Combined with Other Agents

The combination of two ICIs has been recently tested in patients with RCC. Motzer and colleagues [25] recently reported results of CheckMate 214, which is a phase III trial that demonstrated the superiority of the combination of ipilimumab and nivolumab over sunitinib in previously untreated patients with intermediate/poor risk (according to International Metastatic RCC Database Consortium [IMDC] prognostic model) metastatic or locally advanced clear cell RCC (Table 3). This open-label multi-center trial randomized nivolumab (3 mg/kg) + ipilimumab (1 mg/kg) every two weeks vs. sunitinib (50 mg daily) orally once daily for 4 weeks (6-week cycle) as first-line treatment in patients with metastatic clear cell RCC. The coprimary end points were OS, ORR, and PFS. A total of 1096 patients were assigned to receive nivolumab plus ipilimumab (550 patients) or sunitinib (546 patients). Overall, 425 had intermediate risk and 422 had poor risk. At a median follow-up of 25.2 months, the 18-month OS in the intermediate and poor-risk patients was 75% with nivolumab + ipilimumab and it was 60% with sunitinib. The median OS was not reached with the combination therapy vs. 26.0 months with sunitinib (HR, 0.63, *p* < 0.001). ORR was also superior in the ICI group than in the sunitinib group (42% vs. 27%, respectively, *p* < 0.001) and the CR rate was 9% in the combination immunotherapy arm vs. 1% in the sunitinib arm. DCR was similar in the two arms including 72% in the combination group vs. 71% in the control group. PFS was 11.6 months vs. 8.4 months, respectively. With extended follow-up [26] (median follow-up 32.4 months (IQR 13.4–36.3)), results for the three co-primary end points showed that nivolumab plus ipilimumab continued to be superior to sunitinib in terms of OS (median not reached [95% CI 35.6–not estimable] vs. 26.6 months (22.1–33.4). HR 0.66 (95% CI 0.54–0.80), *p* < 0.0001), PFS (median 8.2 months (95% CI 6.9–10.0) vs. 8.3 months (7.0–8.8), HR 0.77 (95% CI 0.65–0.90), *p* = 0.0014), and the ORR (178 (42%)of 425 vs. 124 (29%) of 422, *p* = 0.0001). PD-L1 status was not predictive of response to the combination therapy. Treatment-related AEs occurred in 93% of patients in the combination group and 97% in the control group. G3 or G4 AEs occurred in 46% and 63% of patients, respectively (Table 4). Treatment-related AEs leading to discontinuation occurred in 22% and 12%, respectively. The most commonly reported G3 or G4 AEs in the combination group were elevated lipase levels, fatigue, and diarrhea, while, in the control group, were hypertension, fatigue, palmar-plantar erythrodysesthesia, and elevated lipase levels. There were eight treatment-related deaths in the combination group and four in the sunitinib group.

KEYNOTE-426 was an open-label phase 3 trial that randomized 861 patients with previously untreated metastatic clear cell RCC to receive pembrolizumab (200 mg) IV once every 3 weeks plus axitinib (5 mg) orally twice daily (432 patients) or sunitinib (50 mg) orally once daily for the first 4 weeks of each 6-week cycle (429 patients) [27]. Primary end points were OS and PFS in the intention-to-treat (ITT) population, while secondary end point was ORR. Rationale for choosing axitinib was the best toxicity profile in association with pembrolizumab due to greater selectivity in inhibiting VEGF receptor (VEGFR). Median OS was not reached, but, after a median follow-up of 12.8 months, the estimated percentage of patients who were alive at 12 months was 89.9% vs. 78.3% (HR 0.53, 95% CI, 0.38 to 0.74, *p* < 0.0001). The risk of death was 47% lower with combination therapy when compared to sunitinib. Median PFS was significantly longer with the combination therapy than with sunitinib (15.1 vs. 11.1 months HR 0.69, 95% CI, 0.57 to 0.84, *p* < 0.001). OS and PFS benefit were independent of IMDC risk groups and PDL1 status. The ORR was higher in the combination group than in the control group [59.3% (95% CI, 54.5 to 63.9) vs. 35.7% (95% CI, 31.1 to 40.4), *p* < 0.001]. G3 or G4 AEs of any cause occurred in 75.8% of patients in the combination group and in 70.6% in the control group. In the combination group, AEs of any cause led to discontinuation of either drugs in 30.5% of patients, discontinuation of both drugs in 10.7%, interruption of either drug in 69.9%, and dose axitinib reduction in 20.3%. In the sunitinib group, AEs led to discontinuation in 13.9% of patients, interruption in 49.9%, and dose reduction in 30.1%. The most commonly reported G3 or G4 treatment-related AEs in both groups were diarrhea and hypertension. The incidence of hepatic toxicity was higher in the pembrolizumab-axitinib group. However, there were no deaths related to hepatoxicity. There were four treatment-related deaths in the combination group and seven treatment-related deaths in the sunitinib group. It emerged that Pembrolizumab + Axitinib provides benefits in the combined population of patients with IMDC intermediate or poor risk and in patients whose tumors had sarcomatoid features. The observed benefits were consistent with those seen in the total population. In 2019, Rini et al. presented the analysis of intermediate and poor-risk groups in addition to the sarcomatoid group [28]. 105 patients with sarcomatoid features were examined and results showed improvement in OS (HR 0.58, 95% CI 0.21–1.59, 12-mo rate 83.4% vs. 79.5%), PFS (HR 0.54, 95% CI 0.29–1.00, median not reached vs. 8.4 months), and ORR (58.8% (95% CI 44.2–72.4) vs. 31.5% (19.5–45.6)). Additionally, CR rates were 11.8% (95% CI 4.4–23.9) vs. 0% (0.0–6.6), respectively. 

JAVELIN Renal 101 was a randomized (1:1) phase 3 trial, which evaluated the efficacy of avelumab (10 mg/kg, IV every 2 weeks) plus axitinib (5 mg orally twice daily) or sunitinib (50 mg orally once daily for 4 weeks, in 6-week cycle) in 886 patients with previously untreated advanced RCC patients. OS and PFS in patients with PD-L1–positive tumors were the two independent primary end points. PFS in the overall population, ORR, and safety were secondary end points. A total of 560 (63.2%) patients had a PD-L1-positive tumor. Among these patients, median PFS was 13.8 months in the combination therapy group and 7.2 months in the sunitinib group (HR 0.61, 95% CI, 0.47 to 0.79, *p* < 0.001) in the overall population. Median PFS was 13.8 and 8.4 months, respectively (HR 0.69, 95% CI, 0.56 to 0.84, *p* < 0.001). Among patients with PD-L1-positive tumors, ORR was 55.2% inavelumab plus axitinib group vs. 25.5% in the control group. At the time of the analysis, OS data were immature. G3 or higher AEs were observed in 71.2% and 71.5% of patients, respectively. The most commonly reported G3 or G4 treatment-related AEs were hypertension, diarrhea, increased alanine aminotransferase level, and palmar-plantar erythrodysesthesia in the combination arm, and hypertension, palmar-plantar erythrodysesthesia, and hematological toxicity in the sunitinib arm with three and one treatment-related deaths, respectively [29].

IMmotion151 was a multi-centre, open-label, phase 3, randomized (1:1) controlled trial evaluating the efficacy of atezolizumab (1200 mg) plus bevacizumab (15 mg/kg) IV every 3 weeks or sunitinib (50 mg orally once daily for 4 weeks on, 2 weeks off) as first-line therapy of advanced RCC in patients with a component of clear cell or sarcomatoid histology [30]. Co-primary end points were investigator-assessed PFS in the PD-L1-positive population and OS in the ITT population. Secondary end points were OS in the PD-L1-positive population, PFS in the ITT population, ORR, duration of response, patient-reported outcomes, and safety. A total of 915 patients were enrolled including 40% of whom had PD-L1-positive tumors. Among patients with PD-L1-positive tumors, the median PFS was 11.2 months in the combination arm vs. 7.7 months in the control arm (HR 0.74 95% CI 0.57–0.96, *p* = 0.0217). OS was immature at the time of interim analysis. Overall, 40% of patients in the combination arm and 54% in the sunitinib arm had treatment-related G3–G4 AEs with discontinuation rated at 5% and 8%, respectively. The most commonly reported G3–G4 treatment-related AE in the combination arm was hypertension, while, in the sunitinib group, were hypertension, thrombocytopenia, and palmar-plantar erythrodysaesthesia. There were five treatment-related deaths in the combination group and one treatment-related death in the sunitinib group.

## 5. Anti-PD1/PD-L1 in Non-Clear Cell RCC (nccRCC) and Rare Histological Subtype

Despite the favourable results obtained in terms of improvement in OS from ICI therapy in patients with ccRCC, the efficacy of anti-PD-1/PD-L1 agents in patients with ncc- and rare histological subtype-RCC is still debated. These patient populations are often underrepresented or excluded from clinical trials. Therefore, the relatively small size of those who received ICI therapies precludes a precise estimate of the real benefit provided by anti-PD-1/PD-L1 agents in these subgroups.

In a retrospective case series of non-clear cell RCC (nccRCC) patients, 82% of whom received at least one previous systemic treatment, nivolumab demonstrated an encouraging clinical activity. In particular, DCR was 49%, ORR was 20%, and median PFS was 3.5 months [31]. Similarly, in another retrospective cohort of patients treated with anti-PD-1/PD-L1 agents for a ncc RCC, ORR was 19% and DCR was 52% with median time-to-treatment failure of 4.0 and OS of 12.9 months [32]. In a single-institution experience, 40 patients with metastatic nccRCC received nivolumab as secondary or further line of treatment. ORR and DCR were 20.6% and 70.5%, respectively. Three patients (8.8%) obtained a CR. Median PFS and OS were 4.9 months and 21.7 months, respectively [20]. Consistent with these results, a phase II trial including previously untreated patients with nccRCC showed a remarkable activity of pembrolizumab. In this study, 24.8% of patients had a PR or a CR, 81.5% of whom had a duration of response greater than 6 months. Overall, the rates of PFS and OS at 12 months were 22.8% and 72.0%, respectively [33]. Updated results at a median follow-up of 15 months showed, among patients with the non-clear cell variant, an ORR of 26.1% for the whole cohort and median duration of response of 15.3 months. The predominant histology was papillary (72%) with an ORR of 28%, which was followed by a chromophobe (13%) with an ORR of 9.5%, and unclassified (16%) with an ORR of 30.8%.

Additionally, nivolumab and atezolizumab plus bevacizumab have demonstrated a clinical activity in nccRCC. The initial results from the phase IIIb/IV trial CheckMate 374 showed an ORR of 13.6% and a median OS of 16.3 months. In a phase II trial, the combination of atezolizumab and bevacizumab showed an ORR of 31% and a DCR of 75% [34]. In these studies, patients with unclassified, sarcomatoid or rhabdoid differentiation RCC showed a higher rate of response compared with those observed in patients with chromophobe RCC [35].

More recently, the combination therapy of durvalumab and savolitinib (MET inhibitor) in metastatic papillary renal cancer demonstrated clinical activity with a ORR of 27% and a median OS of 12.3 months [36]. 

Accounting for 1–4% of all adult RCCs and 20–40% of all pediatric RCCs, Xp11 translocation RCC (tRCC) was described as a distinct RCC subtype in the 2004 World Health Organization (WHO) classification, and then it was confirmed as a distinct RCC subtype (i.e., MiT family tRCC) that also includes t(6;11) RCC [37]. The MiT family tRCC harbors the pathognomonic rearrangement of one of the MiT transcription factors TFE3 and TFEB. The prognosis of patients affected by metastatic MiT family tRCC is often poor, since the disease is usually highly aggressive. A retrospective study evaluated the efficacy of ICI therapy in 24 patients with metastatic tRCC. Only one patient received ICI as first-line treatment, while the remaining 23 patients received it as secondary or a further line of treatment. Overall, ORR and DCR were 16.6% and 29.1%, respectively, with median PFS (range, 1–40 months) of 2.5%. Four patients experienced partial response (16.7%) and three patients (12.5%) had stable disease. Despite the limitation of a small number of patients, the study showed objective response in tRCC similar to those observed in ccRCC [38].

## 6. Prognostic and Predictive Biomarkers 

ICIs targeting the PD-1/PD-L1 axis have shown durable responses and clinically meaningful activity in RCC patients. Unfortunately, thus far, there are no reliable biomarkers able to predict the efficacy of immunotherapy and define specific subgroups of patients who best respond to anti-PD-1 agents (Table 5). 

To date, PD-L1 expression is the most studied predictive biomarkers for the outcome of ICI therapy in several cancer types. However, a comprehensive stratification of patient outcomes cannot be predicted using PD-L1 expression due to limited sensitivity, specificity, reproducibility, and poor reliability of the staining. First, PD-L1 expression is highly heterogeneous both within the tissue specimen and between primary tumor and metastatic sites. Second, PD-L1 expression has been reported to be dynamic and deeply influenced by treatments [45,46]. Furthermore, the evaluation of PD-L1 expression is highly dependent on the Ab used. Since different diagnostic antibodies with different staining platforms and scoring systems were used for each anti-PD-1/PD-L1 agent, the reproducibility of immunohistochemical PD-L1 testing is severely limited and sensitivity of each assay cannot be compared to the others [47].

Although the survival benefit associated with ICI therapy was observed across all subgroups of the PD-L1 tumor proportion score (TPS), including PD-L1-negative patients, several studies tried to explore the predictive and prognostic role of PD-L1 in RCC. Baseline PD-L1 expression has been initially associated with poor prognosis in this patient population and data from CheckMate 025 confirmed this correlation, showing a shorter OS in patients with high PD-L1 expression, regardless of what treatment they received (nivolumab or everolimus). 

Although the benefit with nivolumab was observed regardless of PD-L1 expression, among patients treated with nivolumab, median OS was 21.8 months in the PD-L1-positive group when compared with 27.4 months in the PD-L1-negative group [13]. In CheckMate 214 trial, PD-L1 expression has a negative prognostic impact on survival for patients treated with sunitinib but not for those treated with the combination therapy. Nivolumab plus ipilimumab, as the first-line therapy, increased OS and ORR when compared to sunitinib in intermediate-risk and poor-risk patients across all the PD-L1 expression levels, but the difference in benefit was higher in patients with PD-L1-positive tumors than in those with PD-L1-negative tumors. In contrast, PFS benefit was significantly different in patients with a PD-L1-positive tumor but not in those with PD-L1-negative tumors [26]. The analysis of biomarkers from JAVELIN Renal 101 trial showed that PFS is significantly shorter in patients with PD-L1-positive tumors when compared with those who had PD-L1-negative tumors in the sunitinib group. In contrast, no difference in PFS was observed in the axitinib plus avelumab group, according to PD-L1 expression. High-angiogenetic expression signature was significantly associated with better PFS in the sunitinib group, but not in the combination arm. Conversely, PFS was significantly longer in patients treated with axitinib plus avelumab whose tumors were positive for an immune gene expression signature [39]. The negative prognostic effect of PD-L1 expression was initially attributed to its immunosuppressive action. Evidence suggested that T-cell co-stimulation might be implicated in the prognosis of RCC patients. Analyzing B7-H1 expression by immunohistochemistry in patients with RCC who had radical nephrectomy and correlating the results with the survival outcomes, a reduction in the risk of progression and death emerged in B7-H1-negative patients when compared to B7-H1-positive patients. B7-H1 is a co-stimulating glycoprotein in the B7 family that is aberrantly expressed by RCC cells and has been implicated in impairment of T cell function and survival, which resulted in defective host anti-tumoral immunity. In particular, 5-year cancer-specific survival rates were 41.9% in B7-H1-positive patients and 82.9% in B7-H1-negative patients [40]. Moreover, the immunomodulatory role of nivolumab was explored in a hypothesis-generating prospective study in which biomarkers were longitudinally monitored during treatment with nivolumab in a population of mRCC patients. Changes in tumor-associated lymphocytes (e.g., CD3+, CD4+, and CD8+), upregulation of INF-gamma stimulated genes in tumor transcripts, and increase in chemokines (e.g., CXCL9 and CXCL10) were observed, while no significant change in PD-L1 expression under nivolumab treatment was reported [48].

In addition, performing a whole exome sequencing from paired tumor/normal tissue in a cohort of RCC patients who received nivolumab with or without ipilimumab, Miao and colleagues showed an association between PBRM1 loss and survival benefit [41]. Accordingly, PBRM1 loss was identified as a biomarker of response to ICI treatment in an independent cohort of patients from a randomized clinical trial. Patients whose tumors had PBMR1 loss were more likely to obtain a response from nivolumab treatment. Among patients treated with nivolumab, PBRM1 loss was associated with higher ORR, clinical benefit rate, longer PFS, and longer OS. Conversely, among patients treated with everolimus, PFS, OS, and ORR were not different according to the PBRM1 status [42].

Tumor mutational burden (TMB) has been recently proposed as a predictive biomarker of response to ICIs in patients with different cancer types. However, RCC has been shown to have a low mutational load with a median number of non-silent mutations per mega base 10–400 times lower than melanoma or NSCLC [49,50,51]. Despite the low TMB, a high absolute number of indels and an impressive proportion of indels out of total mutations characterize RCC. Indel mutations have been described as the main mechanism responsible for the production of neoantigens to which T-cell responses are directed. Being significantly associated with response to PD-1 agents in melanoma, the high burden of indel mutations seems to be a promising biomarker to predict efficacy of immunotherapy, but the evidence remains controversial [52]. This association was confirmed in a cohort of RCC patients treated with anti-PD-1 agents. Despite the limited sample size, the frameshift indel count, but not the mutational burden, was significantly associated with OS. In contrast, among patients treated with TKI, OS was not statistically different [43].

Recently reported data showing a better clinical response to pembrolizumab (anti-PD-1) in MMR-deficient patients support the hypothesis that MMR-deficient tumors respond better to anti-PD-1 therapy than do MMR-proficient tumors [44].

Data from an exploratory analysis of IMmotion151 trial showed significant improvement in PFS and ORR for patients who received atezolizumab plus bevacizumab whose tumor had a high expression of the T-effector gene signature when compared to those with a low expression [30]. However, neither the T-effector gene signature nor the clinical benefit was associated with TMB, tumor neoantigen burden, indel count, or frameshift mutation burden. 

The high myeloid inflammation gene expression pattern has been supposed to be another biomarker of response to ICI since it was associated with lower PFS in the atezolizumab single-agent group and atezolizumab combined with bevacizumab group, but not in the sunitinib group [44]. 

Additionally, in nccRCC, PD-L1 expression resulted in being a predictive biomarker of poor survival and was associated with more aggressive clinical features such as advanced stage and high grade. According to a retrospective study conducted by Choueiri and colleagues [53], PD-L1 had a wide range of expression among specimens of non-clear RCC. Albeit, the study was limited in sample size. The prevalence of PD-L1 expression in tumor cells was 5% in chromophobe RCC, 10% in papillary RCC, 30% in tRCC, and 20% in collecting duct carcinoma. Analyzing the PD-L1 expression in tumor-infiltrating mononuclear cells, nccRCC showed a marked expression with a prevalence of 36% in chromophobe RCC, 60% in papillary RCC, 90% in Xp11.2 translocation RCC, and 100% in collecting duct carcinoma. Time to recurrence was significantly affected by PD-L1 expression on tumor cells and tumor-infiltrating cells. In particular, patients with PD-L1 expressed in both tumors and tumor-infiltrating cells had a shorter time to recurrence (*p* = 0.02 and *p* = 0.03, respectively) [53].

The molecular characterization could identify additional predictive immunological biomarkers. In a subset of eight patients affected by metastatic MiT family tRCC treated with ICI, the tumor genome was assessed with whole exome sequencing in 4 and targeted sequencing in 4. The median mutational load was lower than that observed in clear-cell RCCs in The Cancer Genome Atlas (TCGA) project [51]. Additionally, mutations in bromodomain-containing genes (PBRM1 and BRD8) were observed in two patients with a long-lasting control of the disease [38].

## 7. Current Treatment Strategies and Perspectives

The favorable results from phase III trials make ICIs combined with anti-CTLA-4 MoAbs or anti-angiogenic agents the preferred first-line of treatment for the majority of patients with RCC. Unfortunately, the clinical trials responsible for the approval of these novel combination strategies have some limitations due to missing information on randomization, unclear interpretation of PD-L1 expression, heterogeneous populations, and absence of study data in data analyses. All these aspects make it difficult to obtain a precise estimation of bias and quality of evidence [54].

Among patients of intermediate or poor IMDC risk, cabozantinib (multi-kinase inhibitor of MET, AXL, RET, and VEGFR2) demonstrated a significant benefit in PFS and ORR over sunitinib as the first-line treatment and is considered one of the standard first-line options [55]. However, the sample size of this phase II study was relatively limited and, when it came to subgroups and survival analysis, could affect the strength of the results. Combination strategies may overcome the resistance to immunotherapy through the immunomodulatory properties of VEGF-TKI inhibitors such as the increase in effector T-cell infiltration and decrease in inhibitory cell subpopulations, which could turn the cold tumors hot. In this setting, cabozantinib seems a promising partner for ICIs and is actually under investigation in clinical trials [56,57,58].

Several issues need to be addressed to define the better treatment strategy, particularly in patients with favorable IMDC risk, non-clear cell, and rare histological subtype. Lastly, further studies are expected to obtain a better selection of patient populations who are more likely to benefit from combination therapies of ICI with anti-CTLA-4 MoAbs or anti-angiogenic agents.

## 8. Conclusions

ICI studies challenge a paradigm shift in the management of RCC. To date, ICIs and targeted therapies are indicated for patients who have a specific risk score and not a specific tumor biomarker. This is consistent with the evidence that, to date, no reliable biomarker is able to predict which patients’ subgroups will benefit from immunotherapy. Another relevant consideration is the definition of ideal time during treatment to administer an ICIs agent. In RCC, although cytoreductive nephrectomy is common, approximately 20–30% of metastasis-free patients will develop recurrent cancer after surgery, which highlights the need for an effective adjuvant therapy. Adjuvant immunotherapy is an attractive approach because it may enable the destruction of the microscopic tumor remaining after surgery, which reduces the relapse risk. According to recent reports, TKIs have not yet proven to be effective in the adjuvant setting, which suggests that a change paradigm of care like immunotherapy could have a role in the high-risk RCC adjuvant setting. 

At last, emerging data with ICIs’ agents and novel combination strategies represents a revolution for managing RCC, which results in an evolving scenario and is likely to further impact clinical decision-making. The establishment of valid predictors of treatment response to ICIs options is required to identify those patients who could benefit from these agents. Furthermore, a single biomarker for patient selection may not be feasible, given that immune responses are dynamic and evolve over time and, hence, will be necessary to integrate multiple components like PD-L1 expression, TILs, mutational load, and many other emergent biomarkers.

## Figures and Tables

**Table 1 ijms-21-04691-t001:** Summary of the main phase II and III trials of anti-PD-1/PD-L1 MoAb single agent.

Trial	Phase	Treatment Setting	Arms	Number of Patients	Risk Groups (%)	Previous Nephrectomy (%)	Primary End-Point	ORR (%)	DCR (%)	PFS (Months)	OS (Months)
NCT0135443/	II	at least 2nd line in mRCC with a clear-cell component after previous antiangiogenic therapy	Nivolumab 0.3 mg/kg, 2 mg/kg, or 10 mg/kg IV every 3 weeks	168	MSKCC risk group: Favorable 33%, Intermediate 42%, Poor 25%	98% of patients received prior surgery (type of surgery not specified)	PFS	20%, 22%, and 20% in each arm respectively	NR	2.7, 4.0, and 4.2 months in each arm respectively	18.2, 25.5 and 24.7 months in each arm respectively
NCT01668784/ CheckMate 025	III	at least 2nd line in advanced or mRCC with a clear-cell component and previous treatment with one or two antiangiogenic therapies	Nivolumab 3 mg/kg IV every 2 weeks vs. everolimus daily oral dose of 10 mg	821	MSKCC risk group: Favorable 36%, Intermediate 49%, Poor 15%	88% of patients received prior nephrectomy	OS	25% with nivolumab and 5% with everolimus	NR	4.6 months with nivolumab and 4.4 months with everolimus	25.0 vs. 19.6
NCT03126331/	II	mRCC of any histology (clear cell 93%, papillary 7%) who received at least one prior anti-angiogenic therapy	Single-arm: nivolumab for 12 w (240 mg every 2 w or 480 mg every 4 w), patients with ≥10% reduction in tumor burden entered a treatment-free observation phase	14	IMDC risk group: Favorable 7%, Intermediate 86%, Poor 7%	100% of patients had a prior nephrectomy	feasibility of intermittent nivolumab (≥80% of patients eligible for intermittent therapy elect to receive intermittent nivolumab)	29%	NR	7.97 months	NR

Abbreviations: MoAb: monoclonal antibody; NR: not reported; w = week.

**Table 2 ijms-21-04691-t002:** AEs in the main phase II and III trials of anti-PD-1/PD-L1 MoAb single agents.

Trial	G3–4 AEs (%)	Most Commonly Reported G3–4 AEs	Discontinuation Rate Due to Treatment-Related AEs (%)
NCT01354431/	11%	Transaminases increased	7%
NCT01668784/CheckMate 025	19% with nivolumab 37% with everolimus	fatigue (3%), anemia (2%), pneumonitis (2%) with nivolumab; anemia (8%), hypertriglyceridemia (5%), hyperglycemia (4%), stomatitis (4%) with everolimus	8% with nivolumab, 13% with everolimus

Abbreviations: AE: adverse event; G: grade; MoAb: monoclonal antibody.

**Table 3 ijms-21-04691-t003:** Summary of the main phase II and III trials of anti-PD-1/PD-L1 MoAbs in combination with other agents.

Trial	Phase	Treatment Setting	Arms	Number of Patients	Previous Nephrectomy (%)	Primary End-Points	ORR (%)	DCR(%)	PFS (Months)	OS (Months)
Checkmate 214	3	First line	Intermediate and poor risk: nivolumab + ipilimumab vs. sunitinib	1096	82% vs. 80%	OS, ORR, PFS	42% vs. 27%	72% vs. 71%	11.6 vs. 8.4	NR vs. 26
KEYNOTE-426	3	First-line	Pembrolizumab + axitinib vs. sunitinib	861	82.6 vs. 83.4	OS, PFS	59.3 vs. 35.7	83.3 vs. 75.1	15.1 vs. 11.1	NR
JAVELIN Renal 101	3	First-line	Avelumab + axitinib vs. sunitinib	886	86.3 vs. 86.9	OS and PFS in PDL1 < 1%	55.2% vs. 25.5%		13.8 vs. 7.2	NR
IMmotion151	3	First-line	Atezolizumab + bevacizumab vs. sunitninb	915 (454 + 461)	84% vs. 83% in PDL1+	PFS in PDL1+ and OS in ITT	43% vs. 34% in PDL1+ 36% vs. 32% in ITT	75% vs. 69% in PDL1+ 75% vs. 72% in ITT	11.2 vs. 7.7 in PDL1+	NR

Abbreviations: MoAb: monoclonal antibody; NR: not reported.

**Table 4 ijms-21-04691-t004:** AEs in the main phase II and III trials of anti-PD-1/PD-L1 MoAb in combination with other agents.

Trial	G3–4 AEs (%)	Most Commonly Reported G3–4 AEs	Discontinuation Rate Due to Treatment-Related AEs (%)
*Checkmate 214*	43 vs. 63	- increased lipase level, fatigue diarrhea - hypertension, palmar-plantar erythrodysesthesia, fatigue, increased lipase level	22 vs. 12
*KEYNOTE-426*	75.8 vs. 70.6	- Diarrhea, hypertension and hepatic toxicity - Diarrhea, hypertension	both drugs: 30.5, sunitinib: 13.9
*JAVELIN Renal 101*	71.2 vs. 71.5	- hypertension, diarrhea, increased alanine aminotransferase level, palmar-plantar erythrodysesthesia - hypertension, palmar-plantar erythrodysesthesia, hematological toxicity	7.6 vs. 13.4
*IMmotion151*	40 vs. 54	- hypertension, - hypertension, thrombocytopenia, palmar-plantar erythrodysesthesia	5 vs. 8

Abbreviations: AE: adverse event; G: grade; MoAb: monoclonal antibody.

**Table 5 ijms-21-04691-t005:** Summary of the main prognostic and predictive biomarkers for immunotherapy in RCC.

Biomarker	Clinical Significance	References
*Immune gene expression signature*	Better PFS among patients treated with axitinib plus avelumab	[39]
*B7-H1 expression*	Negative prognostic factor for 5-year PFS and 5-year cancer-specific survival rate:	[40]
*PBRM1 loss of function*	Better clinical benefit among patients who received nivolumab with or without ipilimumab. Higher ORR, clinical benefit rate, longer PFS, and OS among patients treated with nivolumab.	[41,42]
*PD-L1 expression*	Negative prognostic factor. CheckMate 025 trial: median OS was 21.8 months in patients whose tumors were PD-L1-positive compared with 27.4 months in those whose tumors were PD-L1-negative. However the benefit offered by nivolumab was observed regardless of PD-L1 expression. CheckMate 214 trial: PD-L1 expression had a negative prognostic impact on survival for patients treated with sunitinib but not for those treated with nivolumab plus ipilimumab.	[13,25]
*Frameshift indel count*	Positive predictive factor in patients treated with anti-PD-1 agents.	[43]
*High expression of T-effector gene signature*	Better PFS and ORR among patients treated with atezolizumab plus bevacizumab.	[44]
*High myeloid inflammation gene expression pattern*	Negative prognostic factor for PFS in patients treated with atezolizumab single-agent or atezolizumab combined with bevacizumab	[44]
*Mutations in bromodomain-containing genes*	Long-lasting clinical benefit in two patients with tRCC treated with ICIs.	[38]

Abbreviations: ICI: immune checkpoint inhibitor; ORR: overall response rate; OS: overall survival; PD-L1: programmed death-ligand 1; PD-1: Programmed cell death-1; PFS: progression-free survival; tRCC: translocation renal cell carcinoma.

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
