# Peer review of "Immune Checkpoint Inhibitors in the Treatment of Renal Cancer: Current State and Future Perspective"

_ijms, 2020, doi:10.3390/ijms21134691_

Round 1
Reviewer 1 Report
The authors present a well written and extensive review of immunotherapy in RCC. The main clinical trials, and some interesting and not so well known studies, are discussed. There are some aspects that can be improved, and I think that with those changes it would be a useful paper that allows a fast and complete understanding of the current state of this clinical context.
This are my main suggestions and corrections:
Title: Prospective (maybe "perspective"?)
42. The study by Oliver et al is not comparative, so an increase in OS cannot be confirmed.
47. The phrase is incomplete, or the beggining by "Given..." does not seem correct
51. It is not clear if the authors are referring to anti-PD1 combined with TKIs, or TKIs monotherapy. Also, the order in which the different combinations were approved would clarify this paragraph.
56. ICIs do not neccesarily reactivate immune response, this implies that a response happened before. "Unleash" or "induce" is more appropiate.
138. The aclaration "after nephrectomy" is not needed.
156. Is there any information about Cohort B?
162. RCC is repeated.
162. Which criteria were used to define high risk of recurrence?
Comparations of numerical variables are presented heterogeneously throughout the text. p values are often not reported and they should, while sometimes 95% CI are indicated and they might not add valuable information to the text and difficult the reading (p.e. 191-193)
210. Some discussion of the trials, besides the numerical results, might add value to the paper. Why Axitinib was combined with ICI and not Sunitinib, current SoC? Are outcomes in sarcomatoid tumors comparable to non-sarcomatoid? Which consider the authors reasonable treatment options according to current evidence?
283. There is any data specific to each non-clear cell histology?
287. All the cohort is "non-clear cell variant", so I understand ORR is 26.1% for the whole cohort.
308. Low PFS and high OS. Were subsequent therapies effective? Or is DoR extremely long?
339. If PD-L1 expression only has a negative prognostic value it should affect all the patients, not only those not treated. It might have positive predictive value for ICI treatment, and negative prognostic value.
359. Is it possible that PFS value is higher than cancer-specific survival?
368. WES if performed from which tissue? Primary, metastases?
386. Paragraph not needed.
397. Even - Also?
399. How affected PD-L1 expression on TC/TIC to TTR?
Table 1: The structure is confuse, the columns are not clearly separated. p values should be included when numeric values are compared.
Table 3: If median OS is not reached, please include 6-month of 12-month OS to report informative data.
Table 4: The discontinuation rate of KEYNOTE-426 does not match with the results presented in the text.
Author Response
Reviewer 1
Title: Prospective (maybe "perspective"?)
We have edited as suggested by the reviewer.
- The study by Oliver et al is not comparative, so an increase in OS cannot be confirmed.
We agree with the reviewer’s criticism and we changed accordingly.
- The phrase is incomplete, or the beggining by "Given..." does not seem correct
We have rephrased following reviewer suggestion.
- It is not clear if the authors are referring to anti-PD1 combined with TKIs, or TKIs monotherapy. Also, the order in which the different combinations were approved would clarify this paragraph.
We redraw the sentence following reviewer suggestion.
- ICIs do not neccesarily reactivate immune response, this implies that a response happened before. "Unleash" or "induce" is more appropiate.
We have revised the sentence.
- The aclaration "after nephrectomy" is not needed.
It has been removed.
- Is there any information about Cohort B?
Information regarding cohort B have been introduced in the manuscript following reviewer suggestion.
- RCC is repeated.
It has been removed.
- Which criteria were used to define high risk of recurrence?
The criteria used to define high risk of recurrence have been added in the text and are the following: T2 grade 4, T3a grade 3-4, T3b/c any grade, T4 any grade or TxN+ any grade.
Comparations of numerical variables are presented heterogeneously throughout the text. p values are often not reported and they should, while sometimes 95% CI are indicated and they might not add valuable information to the text and difficult the reading (p.e. 191-193)
We have introduced p value in the revised manuscript.
- Some discussion of the trials, besides the numerical results, might add value to the paper. Why Axitinib was combined with ICI and not Sunitinib, current SoC? Are outcomes in sarcomatoid tumors comparable to non-sarcomatoid? Which consider the authors reasonable treatment options according to current evidence?
We have added in the revised manuscript discussion regarding data of trials. Rationale for choosing axitinib was the best toxicity profile in association with pembrolizumab due to greater selectivity in inhibiting VEGFR. Pembrolizumab + axitinib provides benefit in the combined population of patients with IMDC intermediate or poor risk and in patients whose tumors had sarcomatoid features. The observed benefits were consistent with those seen in the total population
- There is any data specific to each non-clear cell histology?
No data were available on PD-L1 status for ncc histology.
- All the cohort is "non-clear cell variant", so I understand ORR is 26.1% for the whole cohort.
We have revised the sentence in the revised manuscript. The ORR was 26.1% for the whole cohort (n=165 patients), with 43 patients continuing treatment at time of data cutoff. Responses were achieved across all subgroups, including patients with papillary (28.0%), chromophobe (9.5%) and unclassified (30.8%) histologies, or sarcomatoid features (42.1%), as well as for those with an IMDC risk category of favorable (32.1%) or intermediate/poor (23.2%). The ORR was 10.3% for patients with PD-L1 < 1 versus 35.3% for those >1 or more.
- Low PFS and high OS. Were subsequent therapies effective? Or is DoR extremely long?
Due to the small size of the cohort, to the matter it is a retrospective study and additionally enrolling patients heterogeneously treated, it 's not possible to explain the great difference between OS and PFS (range 1-40 months) and to obtain conclusive data. We have revised the sentence.
- If PD-L1 expression only has a negative prognostic value it should affect all the patients, not only those not treated. It might have positive predictive value for ICI treatment, and negative prognostic value.
- Is it possible that PFS value is higher than cancer-specific survival?
We have maintained only cancer-specific survival rate, since PFS analysis was performed in a subgroup.
- WES if performed from which tissue? Primary, metastases?
We have revised the sentence.
- Paragraph not needed.
Unfortunately we have not been able to understand which paragraph is not needed. Please can you please explain again which paragraph? we apologize for the inconvenience.
- Even - Also?
We have revised the sentence.
- How affected PD-L1 expression on TC/TIC to TTR?
We have revised and added the following sentence: “In particular, patients with PD-L1 expressed in both tumors and tumor-infiltrating cells had a shorter time to recurrence (p=0.02 and p=0.03, respectively)”.
Table 1: The structure is confuse, the columns are not clearly separated. p values should be included when numeric values are compared.
We have revised Table 1.
Table 3: If median OS is not reached, please include 6-month of 12-month OS to report informative data.
We have revised the Table.
Table 4: The discontinuation rate of KEYNOTE-426 does not match with the results presented in the text.
We have revised the Table.

Reviewer 2 Report
The study review proposed by Daniele Lavacchi focused on “current state and future prospective of Immune checkpoint inhibitors in the treatment of renal cancer:”
The review is of interest. I have some minor suggestions:
Methodology:
It lacks a description of the objective and analytical elements used to establish the bibliography, the selection criteria and the process of preparing the review
Presentation:
For better visibility and in the form of a figure, also propose one or more diagrams summarising the main points to remember.
As there are no materials and methods, the legends must be explanatory and add some analytical elements of your regressions.
I declare no financial relationships with any organisations that might have an interest in the submitted review; no other relationships or activities that could appear to have influenced the submitted review.
Author Response
Reviewer 2
The study review proposed by Daniele Lavacchi focused on “current state and future prospective of Immune checkpoint inhibitors in the treatment of renal cancer:” The review is of interest. I have some minor suggestions:
Methodology: It lacks a description of the objective and analytical elements used to establish the bibliography, the selection criteria and the process of preparing the review
We have introduced a methodology section according to reviewer suggestion.
Presentation:
For better visibility and in the form of a figure, also propose one or more diagrams summarising the main points to remember. As there are no materials and methods, the legends must be explanatory and add some analytical elements of your regressions.
We have added to the revised manuscript two figures:
- Figure 1. Currently preferred first-line regimens for clear cell histology according to NCCN guidelines (version 2.2020).
- Figure 2. Main phase III trials with anti-PD-1/PD-L1 MoAbs in combination with other agents as first-line treatment.

Reviewer 3 Report
I was very enthusiastic to review this manuscript. While some cochran reviews have come up describing all the clinical studies and proposing guidelines for treatment, there is certainly a gap in the literature for a paper of the scope proposed by the authors in the title and abstract of the article. However, this does not match the main text of the article.
In other words, I found that the title and abstract of this article promised a critical review of the different studies, a discussion of the biomarkers proposed to predict response to immunotherapy in RCC, and a perspective on what could be coming. Thus, I was hoping to find a discussion of the low power of some of the studies reported, some of the deep impacts that these underpowered trials have had in RCC standards of care and some of the current limitations in treatment efficacy, and high patient heterogeneity. Along with an overview of the biomarkers (which by the way are not even described outside of the table), this would have been a great paper I have long wanted to read.
Instead, I found a brief historical overview of immunotherapies for RCC and a description of more recent clinical trials. This has already been described (among other aspects) in recent papers published in 2019, such as:
https://doi.org/10.1186/s40425-019-0813-8
https://doi.org/10.1111/jebm.12362
These papers actually delivered more than a collection of clinical trials for RCC, which found them a great space in the literature. However, this paper as it stands does not add anything to current literature. I feel like those aspects promised in the abstract and title were missed opportunities, and I would very much like to see the authors deliver on those promises.
Author Response
Reviewer 3
I was very enthusiastic to review this manuscript. While some cochran reviews have come up describing all the clinical studies and proposing guidelines for treatment, there is certainly a gap in the literature for a paper of the scope proposed by the authors in the title and abstract of the article. However, this does not match the main text of the article. In other words, I found that the title and abstract of this article promised a critical review of the different studies, a discussion of the biomarkers proposed to predict response to immunotherapy in RCC, and a perspective on what could be coming. Thus, I was hoping to find a discussion of the low power of some of the studies reported, some of the deep impacts that these underpowered trials have had in RCC standards of care and some of the current limitations in treatment efficacy, and high patient heterogeneity. Along with an overview of the biomarkers (which by the way are not even described outside of the table), this would have been a great paper I have long wanted to read. Instead, I found a brief historical overview of immunotherapies for RCC and a description of more recent clinical trials. This has already been described (among other aspects) in recent papers published in 2019, such as:
https://doi.org/10.1186/s40425-019-0813-8
https://doi.org/10.1111/jebm.12362
These papers actually delivered more than a collection of clinical trials for RCC, which found them a great space in the literature. However, this paper as it stands does not add anything to current literature. I feel like those aspects promised in the abstract and title were missed opportunities, and I would very much like to see the authors deliver on those promises.
The therapeutic panorama of renal carcinoma has evolved substantially in the last ten years with a dramatic increase in the number of systemic therapies available. We are now experiencing the era of immunological therapy. Many studies have been done with immunotherapy, alone or in combination, and many others are underway, with unprecedented results in OS. Although this promises to improve patient outcomes, this rapid pace of development has led to new challenges in the therapeutic choice also with regard to the costs and sustainability of health systems. For example, the absence of direct comparative evidence across all treatment options led to a critical gap in the evidence to clearly define the preferred systemic therapy choice. in addition, there is currently no biomarker that can identify the subjects who are most likely to respond to immunotherapy. In this regard, the presentation and discussion made in recent weeks by Dr. Primo Lara at virtual ASCO 2020 (https://www.urotoday.com/conference-highlights/asco-2020/asco-2020-kidney-cancer/121838-asco-2020-biomarkers-of-single-agent-versus-combination-checkpoint-inhibitors-in-advanced-renal-cell-carcinoma.html) sheds light on these issues, ie biomarkers and role in renal cancer. What we know is that those patients with an angiogenic signature are more likely to respond to a combined treatment with VEGFR + TKI. An example is that the combination of pembrolizumab + Axitinib might be more beneficial than NIVO + IPI in patients with a high angiogenesis score. However in our opinion, future validation of this result is required before any recommendation can be made. We strongly believe that direct comparative studies between first-line options linked to predictive biomarkers are needed and that as a new standard in the field, every patient should receive anti-PD-1 therapy as an initial treatment unless there is a specific contraindication to this approach.
Thus concluding, according to the reviewer suggestion we have amended the manuscript: in the revised version there are additional subchapter, Methodogy (subchapter 2) and Current treatment strategies and perspectives (subchapter 6). Furthermore an overview of the biomarkers is reported in subchapter 5 “Prognostic and predictive biomarker”. Moreover we have added to the revised manuscript two figures:
- Figure 1. Currently preferred first-line regimens for clear cell histology according to NCCN guidelines (version 2.2020).
- Figure 2. Main phase III trials with anti-PD-1/PD-L1 MoAbs in combination with other agents as first-line treatment.

Round 2
Reviewer 3 Report
Authors have addressed the concerns presented in the previous review. I believe now this manuscript is more accurate in presenting the reality of RCC treatment.